## RESEARCH ARTICLE

# Significant gap between Point participation and long-term treatment adherence: An evaluation of ivermectin MDA in the Kwanware-Ottou persistent onchocerciasis transmission focus, Wenchi, Ghana

Rogers Nditanchou[1,2*], Akinola Stephen Oluwole[3], Judith Saare[4], David Agyemang[5], Alexandre Chailloux[6], Sandra Adelaide King[5], Mike Yaw Osei-Atweneboana[7], Richard Selby[6], Joseph Opare[4], Anita Jeyam[6], Stephen Pye[6], Louise Hamill[6], Joseph Nelson Siewe Fodjo[2], Elena Schmidt[6], Veronique Verhoeven[2], Robert Colebunders[2]

1 Sightsavers Cameroon Country Office, Yaoundé, Cameroon, 2 Global Health Institute, University of Antwerp, Antwerp, Belgium, 3 Sightsavers Nigeria Country Office, Abuja, Nigeria, 4 Neglected Tropical Diseases Programme, Ghana Health Service, Accra, Ghana, 5 Sightsavers Ghana Country Office, Accra, Ghana, 6 Sightsavers, Haywards Heath, United Kingdom, 7 The Council for Scientific and Industrial Research (CSIR), Accra, Ghana

* rnditanchou@sightsavers.org, nditanchou@yahoo.com

## Abstract

### Background

Despite more than 27 years of ivermectin mass drug administration (MDA), onchocerciasis transmission persists in the Kwanware-Ottou focus within the Wenchi Health District of Ghana. This study examined participation in ivermectin MDA over time in this transmission focus.

### Methods

In March 2024, two months after MDA using the community-directed treatment with ivermectin (CDTI) approach, settlements within Kwanware-Ottou focus were identified through community consultations and satellite imagery. A census was then conducted integrating an ivermectin treatment coverage evaluation survey (CES) to evaluate community participation in CDTI. Data were cleaned using STATA and analysed in R. Descriptive statistics, multiple logistic regression, and ordinal logistic regression were conducted to examine factors associated with point and effective participation in CDTI. Point participation is the percentage of individuals aged 15+ who took ivermectin during the last CDTI, while effective participation refers to those who have taken it at least ten times in past rounds. Pearson correlation was used to assess the relationship between participation and infection prevalence.

**Data availability statement:** All relevant data are within the manuscript and its Supporting information files.

**Funding:** This work received financial support from the Coalition for Operational Research on Neglected Tropical Diseases (COR-NTD), which is funded at The Task Force for Global Health primarily by the Bill & Melinda Gates Foundation (OPP1190754) and by the United States Agency for International Development through its Neglected Tropical Diseases Program. Under the grant conditions of the Foundation, a Creative Commons Attribution 4.0 Generic License has already been assigned to the Author Accepted Manuscript version that might arise from this submission (Grant No 274G to RN). The funders had no role in study design, data collection and analysis, decision to publish, or preparation of the manuscript.

**Competing interests:** The authors have declared that no competing interests exist.

## Results

Nineteen settlements were identified, with an overall point participation of 80.3% (n = 1461 participants; 95% Confidence Interval, CI:78.6 - 82) for the preceding CDTI. However, 10 settlements had coverage below 80%. Effective participation was only 53.5% (n = 974; CI: 51.2 -55.9), well below the recommended 80%. Participation was influenced by factors such as age, occupation, ethnicity, remoteness, length of stay in the settlement, and mobility (migration). Effective participation was correlated with infection levels, with correlation coefficients of -0.74 for microfilariae prevalence and -0.79 for anti-Ov16 seroprevalence, indicating a strong inverse relationship.

## Conclusion

High point participation masks low effective participation and insufficient subdistrict geographical coverage. Conducting exhaustive CES in delineated foci is essential for evaluating CDTI performance, tailoring and strengthening CDTI, and informing alternative strategies to interrupt onchocerciasis transmission. This approach has contributed to effective, context-specific strategies to interrupt transmission in Wenchi and beyond.

## Author summary

Onchocerciasis or river blindness transmission persists in parts of Ghana despite decades of treatment with ivermectin via mass drug administration (MDA). This study focused on the Kwanware-Ottou transmission hotspot, where infection persists despite long-term treatment efforts. Using satellite imagery and settlement consultation, we identified 19 settlements and conducted a comprehensive survey to assess both recent and long-term participation in MDA. We found that while many individuals reported recent treatment during the last MDA round (80.3%), only about half had participated in enough rounds to meet elimination targets. Mobile populations and residents of remote settlements were particularly underserved. Our findings show that high reported coverage during a given MDA round can mask critical gaps in effective MDA participation in the long-term, which are strongly linked to ongoing transmission. We recommend evaluating both point and effective MDA participation in areas with inadequate treatment coverage or persistent transmission. This has contributed to commendable, tailored strategies to interrupt onchocerciasis transmission in Wenchi and beyond.

## Background

Human onchocerciasis is a neglected tropical disease (NTD) caused by the nematode worm *Onchocerca volvulus (O. volvulus)*. It is transmitted to humans through the bite of an infective blackfly of the genus *Simulium*, which breeds in fast-flowing

rivers and streams [1]. The disease burden is traditionally linked to skin conditions, blindness, and the related stigmas [2]. Recent studies increasingly associated onchocerciasis with neurological complications, particularly onchocerciasis-associated epilepsy (OAE) [3]. A preliminary analysis suggests that up to 128,000 years of life lost to disability (YLD) may be due to epilepsy in onchocerciasis-endemic regions of East and Central Africa [3]. This potentially represents over 13% of the total onchocerciasis morbidity burden [3]. Over 240 million people live in endemic areas, 99% of whom are in the Sub-Saharan African Region, where onchocerciasis is endemic in 26 countries [4]. The high burden of onchocerciasis has driven large-scale intervention efforts that continue till today [5–7].

Onchocerciasis control in Africa began with the Onchocerciasis Control Programme in West Africa (OCP), launched in 1974 and concluded in 2002 [8]. The programme initially focused on vector control through aerial larviciding in several West African countries, but from the early 1990s ivermectin delivery became increasingly important, in some areas even replacing vector control [9]. The African Programme for Onchocerciasis Control (APOC, 1995–2015) later continued intervention through community-directed treatment with ivermectin (CDTI) [8]. In Ghana, control efforts under OCP started in 1974 and included vector control, mobile ivermectin treatment, and later CDTI. From 1997, CDTI became the main strategy [5]. The OCP and APOC programmes led to the successful control of onchocerciasis as a public health problem [10,11]. This has spurred the World Health Organization (WHO) to target elimination of transmission (EoT) in at least 12 countries by 2030 [12]. Following APOC's closure in 2015, the Expanded Special Project for Elimination of Neglected Tropical Diseases (ESPEN) now supports countries in achieving elimination targets [13]. Ghana has aligned with this goal [6,8]. However, persistent transmission remains in some areas, such as the Kwanware-Ottou community in Wenchi Health District (HD) [14,15].

The main strategy to achieve EoT is through mass drug administration (MDA) using the CDTI approach [16]. Two key indicators are used to evaluate performance of CDTI: geographic coverage and therapeutic coverage [1,17]. Geographic coverage is defined as the proportion of targeted implementation units in which treatment was delivered—such as districts, communities or settlements [16]. To meet elimination goals, geographic coverage must be sustained at 100% [16]. Therapeutic coverage refers to the proportion of eligible population (age ≥ 5 years) in those areas that receives treatment during each MDA round [18]. A minimum of 80% therapeutic coverage is required to interrupt transmission [18]. For EoT, these two CDTI evaluation thresholds are required to be consistently achieved for at least 12–15 annual rounds of ivermectin MDA [12]. This extended treatment period is necessary because ivermectin is microfilaricidal—it kills only the larvae (microfilariae) of *O. volvulus*, not the adult worms, which can live up to 15 years [19]. Therefore, repeated annual treatments are required to suppress microfilariae levels and gradually reduce transmission until the adult worms die [20].

Operationally, individuals living in onchocerciasis-endemic communities are required to take ivermectin annually for the entire lifespan of the adult worm, which is at least 10 years [20]. To assess this, the WHO recommends that coverage evaluation surveys (CES) be conducted every 1–3 years at the evaluation unit level (often health district) to monitor the effectiveness of CDTI and ensure programmatic coverage targets are met [17,21]. However, programmes struggle to implement CES as often as desired [21]. In addition, reporting coverage of the most recent MDA round in CES gives a false impression of continuous individual participation over the years. Cognisant of this, APOC commissioned a study to evaluate individual continuous participation after eight years (1998–2005) of CDTI [22]. At the time, eight years of treatment was considered the benchmark for transitioning from control to elimination [23]. However, since that study, little attention has been given to evaluating long-term individual adherence to CDTI.

Besides individual-level participation, there is the challenge of evaluating geographical coverage during CES and aligning it with the EoT goal. This is because: firstly, evaluation units often do not align with the transmission focus [15,24]. Secondly, although programmes report community level coverage, geographical coverage is often not assessed during CES. Thirdly, the definition of a community is neither clear nor uniform and there is no agreed approach to assessing programme reports of geographic coverage. Fourthly, endemic areas are dynamic over time and the usual practice of conducting CES at the district level inadequately reflects performance at smaller units. These inadequacies in coverage

in small, remote and newly established settlements, most of the time at high risk of infection [25], are often obscured. Indeed, programmes almost always report 100% geographical coverage (https://espen.afro.who.int/dashboards). These inadequate reports of small settlements coverage [26] pose a significant challenge to EoT [27,28].

To address the dual challenges of evaluating individual participation over the past years and subdistrict geographical coverage, CES need to be adapted particularly where the transmission focus has been delineated. This will align intervention with current onchocerciasis epidemiology and elimination ambition. This paper examines one of such adapted CES. The findings are useful for intervention adjustment and for modellers to review current EoT models.

## Methods

### Ethical consideration

Ethical clearance for this study was obtained from the Ghana Health Service Ethics Review Committee (N° GHS-ERC: 006/09/23). Additionally, administrative authorization was granted by the Director General of Ghana Health Service. Community consultations, mobilization, and sensitization were conducted, during which verbal consent were obtained from community leaders. Informed consent from individuals, and assent for those younger than 18 years, were obtained prior to data collection.

### Study setting

The study was conducted in the Kwanware-Ottou onchocerciasis transmission focus within the Wenchi HD of Ghana in March 2024. Impact evaluations in the Kwanware-Ottou community revealed persistent transmission [15]. A microfilarial (mf) prevalence of 5% (4/72) was found in 2012 and 26.7% (8/30) in 2017 among adults aged 20 years and above. Additionally, an anti-Ov16 seroprevalence of 38.1% (8/21) was found in children aged 5–10 years in the 2017 evaluation, indicating new infections [15]. Table 1 summarizes the documented historical epidemiologic profile of communities within Wenchi HD before 2021.

We conducted a follow-up investigation in 2021 and found a persisting and disturbing mf prevalence of 36.6% (8/24; 95% confidence interval (CI): 19.9% - 56.0%) in Ottou and 29.2% (6/24; CI: 14.6% - 49.8%) in Kwanware. Furthermore, the investigation delineated the transmission focus to be a 10 kilometre (km) radius around Ottou, based on observed infection levels in both human and blackfly [15]. This is consistent with other studies and WHO guidance that suggest a 10–12 km radius as the smallest epidemiological unit for onchocerciasis transmission [1,15,20,29].

**Table 1. Microfilariae (mf) and seroprevalence of onchocerciasis over time in settlements within Wenchi health District [15].**

| Settlements | Anti-Ov16 Seroprevalence[1], % | mf prevalence[2], % | | | | |
|---|---|---|---|---|---|---|
| | 2017 | 1980[3] | 1989 | 2000 | 2012 | 2017 |
| Akete | 2.5 (2/80) | 54.4 (153/281) | – | 2.7(6/221) | – | 1.3 (1/72) |
| Abotereye | – | – | 33.2 (88/265) | – | – | – |
| Adamukura | – | – | 32.2 (38/118) | – | – | – |
| Wurompo | – | – | 23.5 (104/443) | – | – | – |
| Yoyoano | – | – | 20.0 (56/280) | 13.4(15/112) | – | – |
| Kwanware & Ottou | 38.1 (8/21) | – | 48.1(102/212) | 0.7(2/290) | 5.6% (4/72) | 26.7 (8/30) |
| Tainso | | 40.7(127/312) | – | 15.6(17(109) | – | 1.3 (1/72) |

[1]Among children between 5 and 10 years of age; [2]among adult aged ≥20 years; "–" not available/not conducted. [3]Vector control in the Tain River began in 1976 under the OCP, so the prevalence measured in 1980 may not represent the true baseline level. This suggests hyperendemicity and high transmission potential at baseline.

The Kwanware-Ottou transmission focus covers parts of Subsinso, Nsawkaw, and Boase sub-districts in the Wenchi, Tain, and Banda HDs, respectively. See Fig 1. These districts are situated in the Bono Region of Ghana and collectively fall within the Tano-Ankobra Onchocerciasis Operational Transmission Zone (OTZ) [26,30]. Cash crop farming including cashew nuts and fruits is the main occupation in of the people living the area. Additionally, artisanal gold and charcoal mining is a major activity in the Nsawkaw sub-district. Cattle rearing is practiced mainly in the Subinso and Boase areas.

The area is drained by a dense river network, with two main rivers, Subin and Tain, having actual or potential blackfly breeding sites [15]. Vector control began in the Tain River in 1976 under the OCP, and later in the Subin River in 1992 as part of the South-Eastern Extension of OCP—though activities may have started slightly earlier [30,31]. Vector control in both rivers ceased in 1996 [30,31]. Entomological monitoring reveals a significant progress in transmission dynamics over time. At Tainso village near the River Tain, the Annual Biting Rate (ABR) declined from approximately 3,500 bites per person per year in 1974 to fewer than 200 by 1980, with the Annual Transmission Potential (ATP) nearing zero. Subinso village maintained an ABR of approximately 3,000 and an ATP of 150 until 1989, with reductions observed only after 1990 [32].

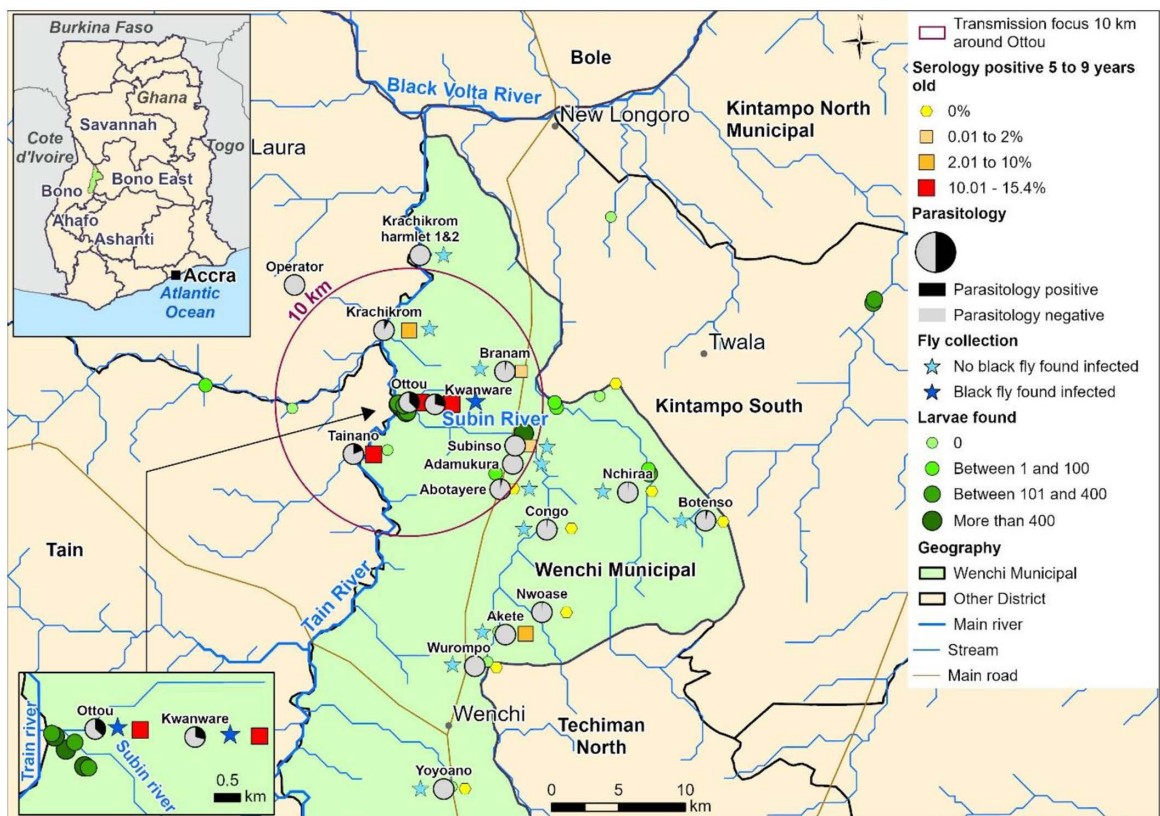

**Fig 1. Map of the study area showing a 10 km radius onchocerciasis transmission focus area around Ottou community.** Parasitology findings, shown as pie charts, indicate the proportion of individuals positive for microfilariae (mf+) detected via skin snip (March 2021). Serology findings are represented by colour-graded squares, with size and shade reflecting Ov16 antibody prevalence (March 2021). Star symbols mark blackfly collection sites, with dark blue stars indicating detection of infective L3 detecting (October 2020). Coloured circles show locations where blackfly larvae were found. The size is proportional to the number of larvae detected. This map has been modified from our previous publication [15]. The base map uses national and district administrative boundaries from the geoBoundaries Global Database (https://www.geoboundaries.org/countryDownloads.html; license: CC BY 4.0, https://www.geoboundaries.org/metadata.html). Additional data sources include HydroATLAS hydrography (https://www.hydrosheds.org/hydroatlas; CC BY 4.0), manually digitized roads from high-resolution satellite imagery, and primary field-collected population and survey data. The map was created in ArcGIS Pro software using only openly licensed datasets.

Annual ivermectin MDA was introduced in 1994 in settlements surrounding both rivers Tain and Subin. Biannual MDA commenced earlier in Tain Districts in 2009 (Banda being part of Tain at the time) and later expanded to Wenchi Municipality in 2018 [15,30,31]. However, like most public health programmes, MDA was suspended during the COVID-19 pandemic, resulting in 2 missed rounds in 2020.

Our 2021 study reported blackfly infectivity rates of 5.9‰ (per thousand) (1/169; 95% CI: 0.1–32.5) in Kwanware and 6.7‰ (2/300; 95% CI: 0.8–23.9) in Ottou, along with the highest blackfly biting rates, ranging from 5,000–6,000 Monthly Biting Rates (MBRs) [15]. These MBRs, though not directly comparable to historical baselines, suggest a rebound in blackfly populations following cessation of vector control. MBRs at this level, especially with infectivity rates above 1‰, can yield ATPs exceeding 100–200 L3/person/year—sufficient to sustain transmission [33]. In such settings, mf prevalence may range from 15% to 40%, particularly where ivermectin coverage is suboptimal or interrupted [1,33]. Indeed, this align with our recent observation in the Kwanware -Ottou focus, as previously noted [15].

## Study design

This was an exploratory study with both cross-sectional and retrospective components. The main outcomes were point (most recent) and past recall of swallowing ivermectin (participation) during CDTI as a function of relevant covariates. The study was nested within a holistic approach aimed at improving CDTI uptake in the 10 km radius area of Kwanware-Ottou focus [15].

## Study procedure

Data collection followed a two-step process. First, all settlements within the defined focus were identified. Then the CES questionnaire (S1 Text) was administered through a door-to-door approach to all households within the focal area. We preferred the term 'settlement' (to community) and defined it as a geographic location where people live. This definition highlights geographical location and considers any physical space, no matter how small.

**Step 1: Identification of settlements through community consultation and Geographic Information (GIS) technology.** GIS technology was used to exhaustively identify settlements in the transmission area. To do this, a recent satellite image of the area, captured in October 2023 with a resolution of 75 cm and covering 271 square kilometres was obtained from Apollo Mapping (https://apollomapping.com/). The satellite image represented 86.3% of the study area. Freely available but low-resolution and old (5 or more years) satellite images of the remaining area were used to complete the coverage. Structures indicative of potential settlements, including mining ones, were identified and mapped based on settlement and mining-related features from previous studies [15,34].

Following the GIS mapping, a series of cascaded consultation meetings, starting from districts through to sub-districts and settlements including hamlets, were held prior to field data collection. During these meetings, dates for settlement-level census and CES were agreed upon with settlement leaders and other members. Additionally, satellite images of the area were presented to settlement dwellers to aid in identification of possible missed neighbouring settlements. Ten trained data collectors verified the mapped potential settlements from 16-24 March 2024. The verified households were grouped as settlements based on their clustering and assigned names as guided by the settlement members. Neighbouring settlements, including for miners, herders, fishermen, and any other groups not initially listed, were searched through inquiry and subsequently included if found.

***Step 2. Census, including a treatment coverage evaluation survey.*** Alongside verification, the team conducted a census one month after the preceding CDTI in all areas identified as settlements based on outcome of consultation and field verification of potential satellite image identified settlements. They also recorded neighbouring settlements and evidence of mining activities and associated miners, including them if not already included. All individuals in the area were targeted for enumeration with relevant variables of interest recorded (see S1 Text).

The primary outcome of interest was individual participation in ivermectin MDA, defined as swallowing ivermectin tablets during CDTI rounds over the past years among individuals aged 15 years and older. The 15-year age cut-off was chosen to ensure that individuals have had the opportunity to participate in at least 10 yearly rounds of CDTI starting from their eligibility age of 5. "Participation" is an inclusive terminology that shifts the responsibility from solely the beneficiary to all stakeholders, including programme implementers and community drug distributors (CDDs). We defined point participation as swallowing ivermectin in the most recent CDTI and cumulative participation as numbers of times an individual had ever participated in CDTI. There are many terminologies used in reference to cumulative participation [35]. Here, cumulative participation refers to what Maddren et al (2023) described as "individual longitudinal compliance", which is how regular an individual had participated in CDTI over time [35].

Since onchocerciasis elimination requires at least 10 years of annual treatment [1,20,36], we defined effective participation as participating in ivermectin MDA at least 10 times as a proxy to 10 consecutive years. While the "≥10 times" participation serves as a proxy for 10 years of treatment in a context of annual MDA, it may not precisely reflect the 10-year duration in our study site due to the shift to biannual treatment from 2018. Any participation less than 10 times was classified as non-effective for the purpose of elimination. Response options for CDTI participation were structured to aid recall as follows: "never or once", "2–4 times", "5–9 times", and "≥10 times". These were later reclassified as "<10" and "≥10 times" corresponding to non-effective and effective participation, respectively. However, each of the shorter participation categories was also explored to estimate the magnitude and direction of the correlation with infection prevalence determined in the 2021 epidemiological survey in the area [15], and to validate the "≥10 times" effective participation cut-off.

Explanatory variables of interest included:

1. Movement: Whether individuals travelled outside the settlement for at least one month in the past 12 months (also used as a proxy for longer past movement).

2. Length of stay in the settlement: The rationale here is that individual's participation and transmission/infection are dependent on their length of stay. This is defined as number of years a participant has been living in the settlement, categorized as "0-5 years", "6-10 years", and "above 10 years". This categorisation aimed to aid recall among participants

3. Settlement: Grouped as main (accessible established settlements) and satellite (often remote and sometimes temporal).

4. Demographic variables: Age, sex, ethnicity, education, and occupation

## Statistical analysis

Survey data was downloaded from the Commcare platform and cleaned using STATA statistical software (StataCorp. 2023. Stata Statistical Software: Release 18. College Station, TX: StataCorp LLC). R version 4.4.1 was subsequently used for analysis [37]. CDTI participation proportions were calculated as percentages. The McNemar test of paired outcomes was used to compare the proportions of point and effective participation.

Before conducting regression analysis, key statistical assumptions were verified. For multicollinearity, variance inflation factors (VIFs) were computed. None of the included variables had a VIF ≥ 5 (all < 2), indicating negligible collinearity [38]. We confirmed that sample counts for each category were within the acceptable range (8–25) [39], ensuring robustness. We also ran a sensitivity analysis by comparing regression models with and without merging a small category (Mining [unauthorised], n = 19) into a similar group (Cattle Rearing). Results were consistent across models, with minimal changes in coefficients and model fit. This supports keeping the categories separate due to their theoretical relevance.

Multiple logistic regression analyses were conducted to estimate the association (odds ratios, OR) between co-variates and point and effective participation in CDTI. Additionally, ordinal outcomes were derived as 0, 1, 2 and 3 corresponding to "Never or once", "2-4", "5-9", and "≥10 times" of participation. Prior to this regression, the proportional odds assumption

was verified using the likelihood ratio test (LRT) ensuring it was met. A multivariable ordinal logistic regression was then conducted to predict the adjusted odds ratio (OR) of change from a lower to a higher category. McFadden's pseudo-$R^2$ statistics was used to compute the percentage variation explained by logistic regression models. P-values less than 0.05 was considered statistically significant.

All participants were included in the logistic regression analysis, regardless of their length of residence in the community. This approach reflects the epidemiological importance of CDTI participation: individuals present in the community contribute to transmission dynamics and are expected to receive treatment. Therefore, participation evaluation was not restricted by duration of residency.

To examine the relationship between CDTI participation and infection prevalence, a dataset was prepared with variables for different level of cumulative participation and infection metrics (mf and anti-Ov16 seroprevalence observed in 2021 across settlements in the study area [15]. Correlation analyses were conducted using the Pearson method to assess the strength and direction of the relationships [40].

## Results

### *Identification of settlements through settlement consultations and Geographic Information (GIS) technology*

Combining GIS and census data, 19 named settlements were identified. Ten of the 19 settlements were identified during settlement consultation. These were remote hamlets with poor accessibility and often merged into bigger settlements for CDTI. See Fig 2 for the map of the settlements which also shows point and effective participation. There were also many unoccupied temporal shelters usually occupied periodically by cattle rearers/herders and artisanal miners during farming and mining season respectively.

### Community-directed treatment with ivermectin (CDTI) coverage

Of the 3,338 individuals surveyed, only those aged ≥15 years who provided valid information on past ivermectin intake were retained to allow a consistent comparison of point and historical participation. Consequently, 1,819 participants were included in the analysis. Their median age was 30 (interquartile range, IQR: 22–45) years, with 929 (51.1%) males and 890 (48.9%) females. In all settlements, effective participation was (almost always significantly) lower than the point participation (Table 2). Overall point participation in the most recent ivermectin MDA was high (80.3%; 95% Confidence Interval, CI:78.6 - 82), but effective participation was substantially lower at 53.5% (95% CI: 51.2, 55.9) (p < 0.001). Ten settlements fell below the 80% recommended point participation threshold, with settlements having remote and mobile populations particularly underserved. More than half of the participants (65.3%), essentially the young population (aged 15–24 years), has an effective coverage of 50.8%. Ottou has very high effective participation (90.2%) which was very close to its point participation (97.6%). The effective participation in Ottou exceeds the upper outlier threshold of 88.2% (defined as mean+/-2 Standard Deviation (SD); mean = 53.6%, SD = 11.7). Thus, effective participation in Ottou is an outlier from the statical standpoint. Without Ottou, the overall effective coverage drops slightly to 52.7%. However, neighbouring Kwanware has a much lower effective coverage (40.4%). The two settlements (Kwanware and Ottou) are located near each other and share similar population characteristics and infection levels [15].

Table 3 presents recent mf and seroprevalence levels obtained in 2021 [15], along with effective participation rates as observed in this study. Meanwhile Fig 3 shows correlation between effective participation and the infection prevalence. A strong negative correlation was observed between effective participation in CDTI and infection prevalence in 2021. For mf prevalence, the correlation coefficient was - 0.74, indicating that settlements with higher effective participation in CDTI experienced significantly lower mf prevalence. Likewise, for anti-Ov16 seroprevalence, the correlation coefficient was - 0.79, demonstrating an even stronger inverse relationship, where low effective participation was associated with higher anti-Ov16 prevalence. In contrast, lower participation categories (2–4 and 0–1 rounds) were positively correlated with infection indices, particularly microfilarial prevalence.

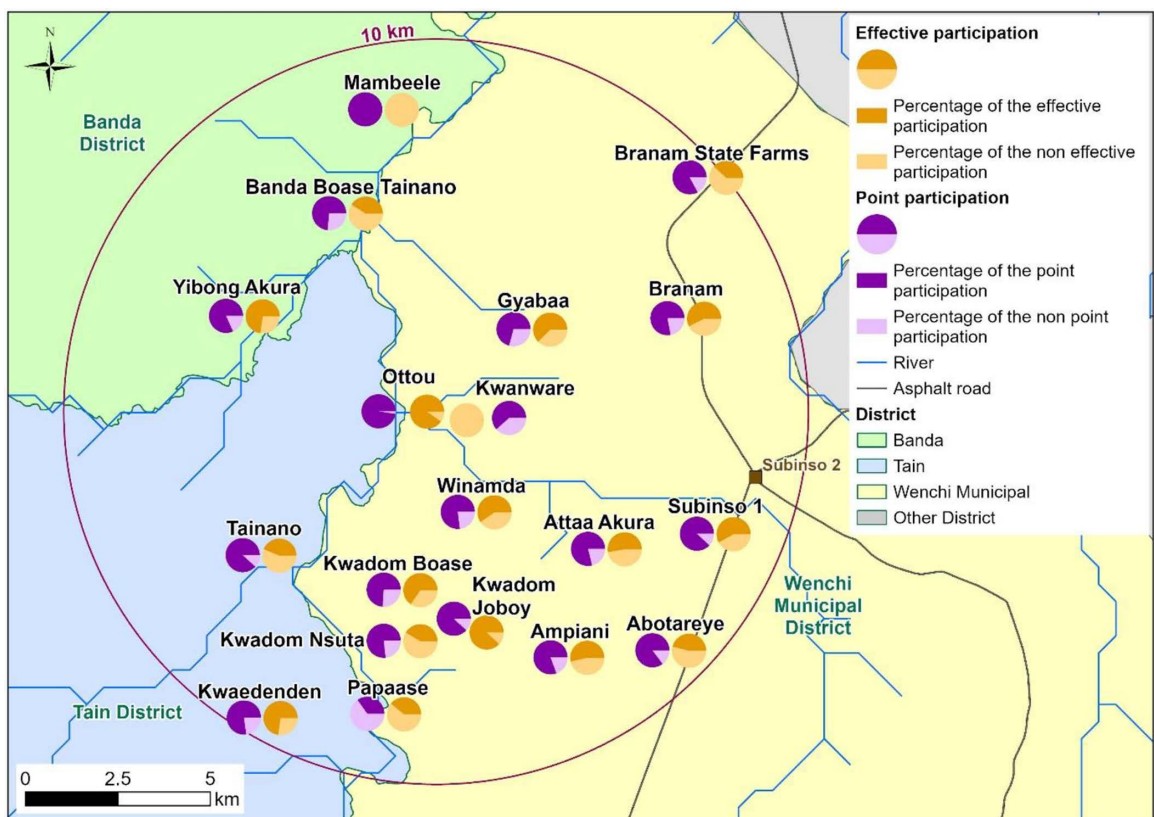

**Fig 2. Map of the study area.** The 10 km radius delimits the focus of persistent infection around Kwanware-Ottou community. Serological and parasitological surveys were conducted in February and March 2021, while entomological surveys—including fly collection, dissection, and breeding site assessment—were carried out in October 2020 [15]. The base map uses national and district administrative boundaries from the geoBoundaries Global Database (https://www.geoboundaries.org/countryDownloads.html; license: CC BY 4.0, https://www.geoboundaries.org/metadata.html). Additional data sources include HydroATLAS hydrography (https://www.hydrosheds.org/hydroatlas; CC BY 4.0), manually digitized roads from high-resolution satellite imagery, and primary field-collected population and survey data. The map was created in ArcGIS Pro software using only openly licensed datasets.

Multivariable ordinal logistic regression model indicates that being of older age slightly but significantly increases the odds of taking ivermectin more frequently (OR = 1.06; 95% CI: 1.05-1.07). In addition, individuals living in Nsawkaw (OR = 2.12; 95% CI:1.31-3.43) and Subinso (OR = 2.12; 95% CI:1.48-3.03) sub-districts, compared to Banda, have significantly higher odds of being in the higher category of ivermectin participation (see Table 4). An individual's length of stay also influences the odds, with shorter stays significantly associated with a lower likelihood of being in a higher category compared to ≥10 years stay category. For ethnicity, those belonging to the Dagaaba and Fulani groups have significantly lower odds of being in a higher category of ivermectin participation with respect to Akan ethnic group. Likewise, miners and students or other occupations have significantly lower odds of being in the higher categories of cumulative participation compared to farmers. The model is most stable for change from 2 (5–9 times) to 3 (≥10 times) cumulative participation categories.

The associations observed for effective and point participation were consistent in direction as the same covariates tended to increase or decrease the odds of participation across both models (See Table 5), though the strength and statistical significance varied. Travel category is an exception. The effective participation model explains a greater proportion of the variation (McFadden's pseudo-R² = 34.2%) suggesting stronger predictive performance in comparison to the point participation model (pseudo-R² = 14.7%).

Table 2. Comparison between effective and point participations by settlement.

| Settlements | Total, n | Effective participation, % | Point participation, % | p-value |
|---|---|---|---|---|
| Overall | 1819 | 53.5 | 80.3 | <0.001 |
| Ampiani | 27 | 44.4 | 96.3 | <0.001 |
| Ata Akura | 19 | 52.6 | 78.9 | 0.07 |
| Boase Tainano | 163 | 41.1 | 73.6 | <0.001 |
| Branam state farms | 47 | 38.3 | 83.0 | <0.001 |
| Gyabaa | 24 | 62.5 | 70.8 | 0.683 |
| Kwadom Boase | 104 | 65.4 | 74.0 | 0.233 |
| Kwadom Joe-boy | 17 | 88.2 | 88.2 | 1 |
| Kwadom Nsuta | 82 | 41.5 | 76.8 | <0.001 |
| Kwaedenden | 23 | 72.7 | 77.3 | 1 |
| Papaase | 23 | 39.1 | 34.8 | 1 |
| Winamda | 52 | 59.6 | 76.9 | 0.095 |
| Yibong Akura | 11 | 72.7 | 81.8 | 1 |
| Abotareye | 232 | 46.1 | 85.3 | <0.001 |
| Branam | 446 | 57.4 | 78.0 | <0.001 |
| Kwanware | 52 | 40.4 | 61.5 | 0.006 |
| Ottou* | 41 | 90.2 | 97.6 | 0.243 |
| Subinso 1 | 353 | 58.1 | 88.1 | <0.001 |
| Tainano | 103 | 43.7 | 88.3 | <0.001 |

*Ottou reported an effective participation rate of 90.2%, which exceeds the upper outlier threshold of 88.2% - define as mean+/-2 Standard Deviation, SD (mean = 53.6%, SD = 11.7).

Table 3. Settlement level onchocerciasis infection prevalence [15] and community-directed treatment with ivermectin (CDTI) effective participation.

| Settlement | Infection indices, % (95% Confidence Intervals), (2021) | | Cumulative participation level, % (2024) | | | |
|---|---|---|---|---|---|---|
| Settlement | Microfilaria | Anti-Ov16 | ≥10 times | 5-9 times | 2-4 times | 0-1 time |
| Ottou | 40.0 (21.4–62.0) | 8.3 (1.2–41.4) | 90.2 (37/41) | 7.3 (3/41) | 0 (0/41) | 2.4 (1/41) |
| Kwanware | 30.0 (14.1–52.7) | 13.3 (1.3–40.6) | 40.4 (21/52) | 19.2 (10/52) | 9.6 (5/52) | 30.8 (16/52) |
| Tainano | 20.0 (11.7–32.0) | 13.0 (4.3–33.6) | 43.7 (45/103) | 26.2 (27/103) | 26.2 (27/103) | 3.9 (4/103) |
| Boase Tainano | 9.0 (4.1–18.5) | 19.0 (6.9–42.8) | 41.1 (67/163) | 26.4 (43/163) | 25.2 (41/163) | 7.4 (12/163) |
| Abotareye | 4.3 (1.6–10.9) | 0 (0–0.8) | 46.1 (107/232) | 37.9 (88/232) | 12.9 (30/232) | 3.0 (7/232) |
| Branam | 3.1 (1.0–9.2) | 1.2 (0.2–7.8) | 57.4 (256/446) | 27.1 (121/446) | 9.9 (44/446) | 5.6 (25/446) |
| Subinso 1 | 0 (0–3.8) | 1.3 (0.2–8.9) | 58.1 (205/353) | 30.9 (109/353) | 9.3 (33/353) | 1.7 (6/353) |

Recent infection levels were available only for seven settlements within the study area [15]. Ottou was excluded from correlation analysis because effective participation rate of 90.2% was determined to be an outlier. Microfilarial prevalence was assessed among adults aged ≥20 years using two skin snips per person, collected from the iliac crests. The snips were examined using classic microscopy to detect the presence of *Onchocerca volvulus* microfilariae. Anti-Ov16 IgG4 seroprevalence was measured in children aged 5–9 years using dried blood spot (DBS) samples, analyzed with the Standard Diagnostics (SD) Ov16 rapid diagnostic test (RDT) to detect IgG4 antibodies specific to *O. volvulus*.

## Discussion

This study comprehensively investigated both cumulative and point CDTI participation through a census in the Kwanware-Ottou onchocerciasis transmission focus within the Wenchi HD of Ghana. The use of settlement knowledge and GIS ensured the inclusion of all settlements within the delineated transmission focus [15]. The exhaustive sampling

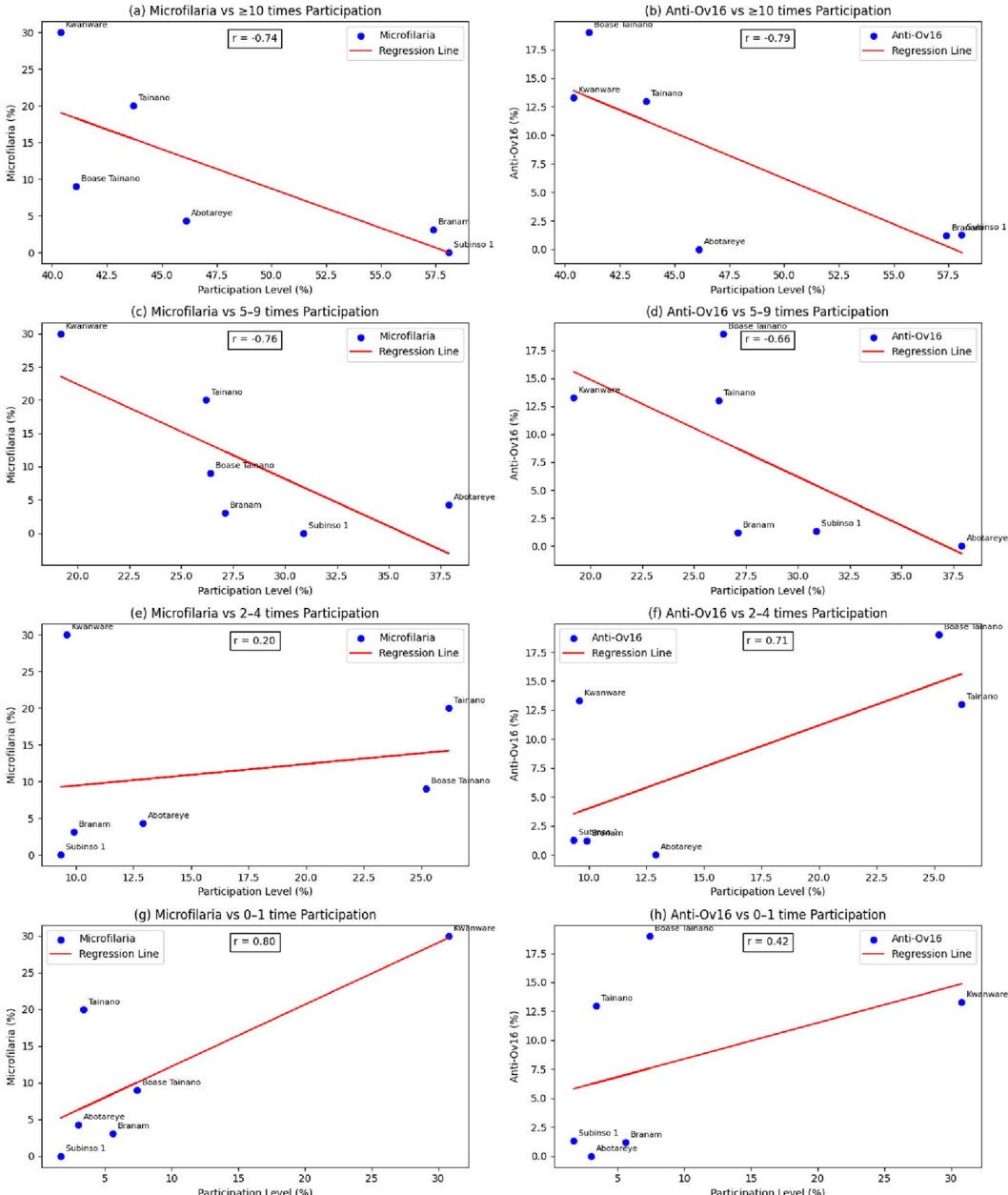

**Fig 3. Microfilaria and Ov-16 Prevalence: Correlation with Ivermectin Participation Level.** (a) Microfilaria vs ≥ 10 times Participation; (b) Ov-16 vs ≥ 10 times Participation; (c) Microfilaria vs 5–9 times Participation; (d) Ov-16 vs 5–9 times Participation; (e) Microfilaria vs 2–4 times Participation; (f) Ov-16 vs 2–4 times Participation; (g) Microfilaria vs 0–1 time Participation; (h) Ov-16 vs 0–1 time Participation.

approach aided accurate and precise estimates of coverage [41]. In addition, this approach provided a detailed understanding of population location and demographic characteristics as part of the investigation into a persistent onchocerciasis transmission focus [15,41]. This aligns with the growing call to accelerate elimination by adapting onchocerciasis interventions to local context [15,20,26,42]. While CES are often conducted at the district level with random sampling,

**Table 4. Multivariate ordinal logistic regression showing the association between frequency of participation in community-directed treatment with ivermectin and covariates.**

| Characteristics | Predictor category | Total, n | Participation % (95% CI) | Adjusted OR (95% CI) |
|---|---|---|---|---|
| Age* | Individual age | 1819 | 80.3 (78.6- 82.0) | 1.06 (1.05 -1.07) |
| Sex | Female | 890 | 79.7 (76.8–81.9) | Reference |
| | Male | 929 | 81.6 (79.0–83.6) | 0.92 (0.75–1.14) |
| Ethnic group** | Akan | 237 | 82.5 (76.1–87.3) | Reference |
| | Dagaaba | 1029 | 78.8 (76.2–81.0) | 0.57 (0.39–0.82) |
| | Fulani | 48 | 68.4 (54.2–79.2) | 0.17 (0.08–0.33) |
| | Mo | 143 | 80.5 (73.1–85.9) | 1.32 (0.79–2.21) |
| | Others | 361 | 86.8 (83.2–89.5) | 0.63 (0.42–0.94) |
| Sub-District | Banda | 175 | 76.5 (69.7–81.5) | Reference |
| | Nsawkaw | 125 | 88.6 (82.6–92.6) | 2.12 (1.31–3.43) |
| | Subinso | 1519 | 80.4 (78.1–82.0) | 2.12 (1.48–3.03) |
| Occupation*** | Farming | 1224 | 80.1 (77.7–81.9) | Reference |
| | Cattle rearing | 60 | 82.5 (73.0–89.0) | 1.35 (0.75–2.41) |
| | Mining (authorised) | 16 | 62.7 (40.5–80.1) | 0.36 (0.13–0.97) |
| | Sale/office/civil servants | 224 | 84.8 (79.5–88.5) | 1.02 (0.73–1.43) |
| | Student/other | 219 | 81.1 (75.4–85.3) | 0.56 (0.40–0.77) |
| | Unemployed | 76 | 78.4 (68.1–85.7) | 0.72 (0.43–1.20) |
| Education | Never attended school | 1052 | 80.4 (77.8–82.6) | Reference |
| | Ever attended school | 767 | 81.0 (77.6–83.5) | 0.78 (0.61–1.0) |
| Settlement category | Main | 1031 | 83.0 (80.1–84.9) | Reference |
| | Satellites | 788 | 77.8 (74.2–80.5) | 0.94 (0.73–1.21) |
| Movement | Not travelled outside district | 1666 | 82.8 (80.8–84.2) | Reference |
| | Travelled outside district | 114 | 63.9 (55.4–71.7) | 1.16 (0.76–1.7) |
| | Visitor | 39 | 47.9 (31.5–65.4) | 0.65 (0.33–1.3) |
| Length of stay | ≥10 years | 1231 | 86.4 (84.0–88.1) | Reference |
| | 0–4 years | 357 | 62.8 (57.2–67.6) | 0.06 (0.04–0.08) |
| | 5–9 years | 231 | 80.3 (74.7–84.4) | 0.23 (0.17–0.3) |

*Age was analysed as a continuous variable. **One missing. ***No participants reported fishing as their occupation. Ordinal scale included: ≥ 10 times, 5–9 times, 2–4 times and 0–1 time. Coverage is descriptive only and not included in the regression.

impact evaluations typically target all individuals in riverine areas. Although this CES approach was appropriate in the early stages of the programme implementation, it now requires adaptation and alignment to impact evaluation thus reflecting the current reality—onchocerciasis transmission has now shrunk to localized hotspot areas.

Although none of the 19 identified settlements had zero-point participation, insufficient participation was reported in 10 of them. Eight settlements had point participation between 70% and 80%, one had 61.5%, and another 34.8%. These coverage estimates warrant cautious interpretation. Because some settlements have very small populations, the exclusion or mistargeting of just a few households can markedly affect the calculated coverage, potentially overstating or understating true participation. This issue is particularly relevant in the context of elimination efforts that aim to ensure no one is left behind.

The settlements with suboptimal point participation were typically remote, located close to known vector breeding sites, inhabited by highly mobile populations and not identified as separate identities in the CDTI register. These factors challenged the already insufficient logistic and financial resources and further constrained the ability to reach the people for

**Table 5. Factors associated with ivermectin participation.**

| Characteristics | Variables | Effective participation Adjusted OR (95% CI) | Point participation Adjusted OR (95% CI) |
|---|---|---|---|
| Sex | Female | Reference | Reference |
| | Male | 1.10 (0.85-1.42) | 1.21(0.94-1.5) |
| Age* | Age | 1.07 (1.06-1.08) | 1.01(1.00-1.02) |
| Ethnic group | Akan | Reference | Reference |
| | Dagaaba | 0.51 (0.33-0.77) | 0.67 (0.39-1.15) |
| | Fulani | 0.08 (0.02-0.29) | 0.33(0.13 - 0.90) |
| | Mo | 1.33 (0.75-2.36) | 1.03 (0.50 -2.11) |
| | Others | 0.60 (0.38-0.96) | 1.2 (0.66 - 2.17) |
| Main occupation | Farming | Reference | Reference |
| | Cattle rearing | 1.86 (0.88-3.90) | 1.11 (0.57 - 2.19) |
| | Mining | 0.39 (0.10-1.51) | 0.26 (0.10 -0.72) |
| | Sale/office/civil servants | 1.13 (0.75-1.69) | 1.60 (1.04 - 2.46) |
| | Student | 0.41 (0.27-0.64) | 0.99 (0.66 -1.48) |
| | Unemployed | 0.67 (0.35-1.31) | 0.70 (0.38 -1.28) |
| Travel | No travel outside district | Reference | Reference |
| | Travelled outside district | 1.35 (0.81-2.25) | 0.26 (0.17 – 0.38) |
| | Visitor | 1.65 (0.54-5.06) | 0.18 (0.08 - 0.40) |
| Duration of stay in the settlement | ≥10 years | Reference | Reference |
| | 0-4 years | 0.07 (0.04-0.10) | 0.23 (0.17 - 0.31) |
| | 5-9 years | 0.15 (0.11-0.22) | 0.63 (0.44 - 0.91) |
| Education | Yes | Reference | Reference |
| | No | 0.91 (0.68-1.22) | 1.04 (0.79 - 1.39) |
| Sub-District | Boase | Reference | Reference |
| | Nsawkaw | 1.92 (1.06-3.49) | 2.55 (1.35 - 4.80) |
| | Subinso | 1.96 (1.28-3.00) | 1.27 (0.83 - 1.94) |
| Settlement category | Main | Reference | Reference |
| | Satellites | 1.09 (0.80-1.49) | 0.70 (0.51 - 0.95) |

*Age was analysed as a continuous variable.

treatment. The settlements with insufficient point participation could have been missed given the 80.3% overall point participation if not of the census approach. This demonstrates that the usual non-exhaustive sampling during CES might hide pockets of insufficient coverage sustaining transmission. Hence, exhaustive sampling CES would be appropriate when conducting evaluation in endemic communities with known persistent transmission like our study area. In the context of onchocerciasis elimination, these are critical locations to reach to clear transmission.

Effective participation was 53.5%, lower than the 80.3% point participation for the most recent MDA in the area and the recommended 80% therapeutic coverage. In addition, more than half of the participants, essentially the dynamic young population, has an effective coverage of less than 55%. A study mandated by APOC in 2011 found a similar result with only 44.9% of individuals having taken ivermectin 6–8 times out of 8 CDTI rounds implemented [22]. The 44.9% in this study was similarly much lower than the preceding MDA reported coverage of 74%. This demonstrates that the high point participation does not reflect individual consistent participation over time. This finding most likely explains the disparity between observed and expected transmission indices including the persistent transmission in the Kwanware Ottou focus [15,43,44].

Evaluating CDTI participation using a 10-round cut-off aligns with both statistical strength and biological plausibility. Statistical robustness was demonstrated by the ordinal multiple logistic regression model, which was most stable when shifting from the '5–9 times' to the '≥10 times' treatment categories. In addition, the multiple logistic regression explained 34.2% of the variation in effective participation—substantially higher than the 14.5% explained by point participation and the 8% reported by Breiger et al [22]. Biological plausibility was supported by correlation analyses, which revealed a strong inverse relationship between cumulative ivermectin intake and onchocerciasis infection. Participation in 5–9 and ≥10 rounds was negatively correlated with microfilariae and anti-Ov16 prevalence, with the strongest effect in the ≥10 times participation group (-0.74 for mf and -0.79 for anti-Ov16 seroprevalence). These findings align with those of Wanji et al in Cameroon, who reported highest mf prevalence (59.7%) in their zero-treatment group and the lowest (33.9%) among those treated seven or more times (OR = 2.8; 95% CI: 2.09–3.74; $P < 0.001$) [45]. The implication is that programmes should incorporate evaluation of cumulative participation during CES and adopt the "10-times" cutoff when assessing effective participation. This strengthens Wanji et al's suggestion that evaluation of cumulative participation (adherence) should be the first step of, or at least conducted together with, impact evaluation since there is correlation between infection and cumulative participation [45].

The correlation analysis equally indicated that participation in five or fewer rounds may falsely suggest consistent and impactful adherence. Practically, if ≥80% of the population has not taken ivermectin at least five times, transmission reduction is unlikely. Put differently, regardless of coverage level, achieving elimination of transmission (EoT) is unrealistic when most individuals have participated in fewer than five rounds. As earlier mention, regression and correlation results support using "<10 times" and "≥10 times" as key thresholds, with 0–4 and 5–9 as intermediary categories. An additional "0 or 0-1 time" group should be added given the growing interest in measuring systematic refusals of ivermectin, i.e., individuals never treated in MDA programmes [46,47].

The categorization of the number of past ivermectin intakes does not review individuals' exposure to MDA nor pattern of interruptions in treatment—whether consecutive or intermittent - which could allow adult worms to resume microfilariae production and sustain transmission. Recently, distribution of ivermectin in Ghana was jeopardized by the suspension of 2 rounds of MDA during 2020 due to COVID-19 pandemic-related restrictions. Although contemporary studies and models did not show a significant impact from brief suspensions of CDTI, the debate is not completely settled, given that it takes 12–24 months for new adult worms to establish and begin producing microfilariae [19,48]. The combination of MDA consecutive or intermittent interruption and low effective coverage may amplify negative impacts on elimination progress in Ghana and beyond [49,50]. There is the need for further investigation to assess the number and consistency of treatments, alongside contextual factors such as blackfly and human population dynamics, to determine the transmission break point. In the context of programmatic monitoring through CES, asking how many times an individual has not taken ivermectin in the last 10 rounds/years can be considered for triangulation and assessment of consecutiveness or intermittency.

Ottou settlement shows both high point and effective participations in ivermectin treatment (90.2% and 97.6%, respectively; Table 2), yet infection levels remain high (40.0%; Table 3). Although statistical analysis identified Ottou's effective coverage as an outlier and it was excluded from the correlation analysis, this finding warrants careful interpretation. Onchocerciasis transmission is highly heterogeneous [51], unlike infections such as influenza where exposure is more uniform [52]. The heterogenicity in Ottou and surrounding is likely due to high biting rates, as previously noted [15], combined with the lower coverage in nearby settlements [15]. Alternatively, reliability of reported participation might have been affected by the very small sample size—only 41 individuals—along with potential biases such as recall bias or conflation between recent CDTI round and those conducted in the more distant past. However, these explanations are not supported by observations in neighbouring Kwanware, which shares similar population characteristics but has much lower effective participation and similarly high infection levels (Table 3). Therefore, it appears that the most likely explanation is conflation and/or biases—most notably social desirability bias [53].

Factors associated with point and effective participation were similar in magnitude and direction, although to varying extents and degrees of statistical significance except for the travel category. This is expected since effective participation is a cumulative evaluation of point participation. Though the individuals may not be the same, the factors may be. Age, working in an office or as a retailer, and living in the Nsawkaw sub-district were associated with higher odds of effective participation. The observed difference for the travel category likely reflects the cumulative nature of effective participation. Individuals who travel frequently may still be present during some MDA rounds and thus report recent (point) participation, but their mobility reduces the likelihood of consistent participation over time. Increasing age means increasing frequency of exposure to MDA and having an occupation that requires less mobility, and migration means a higher likelihood of being met by CDDs. Sub-district differences may be due to different programmatic endeavours at this level, as well as the different characteristics of participants. For instance, the settlers in the Nsawkaw sub-district are mainly farmers, whereas Boase settlers are mainly miners who are equally mobile and additionally avoid people from the government as their activity is illegal [54]. Factors linked to low participation include belonging to the Fulani ethnic group, being miners ('galamsay' -unauthorised), living in satellite settlements, being mobile, and having a short stay in the settlements. This is expected since all these characteristics make it difficult to implement CDTI. The factors influencing effectiveness are most likely lasting meaning a sustained effort is required to address them for a lasting impact. While these factors highlight beneficiary characteristics, aspects of the intervention programme—such as periodicity, reach, equity, and logistics—also warrant attention. A critical issue is determining whether suboptimal coverage in certain settlements is driven more by limited willingness to take ivermectin in that round or by CDDs not reaching all households, either because they moved too quickly or inadvertently missed some homes. As discussed earlier, both mechanisms operate simultaneously, although their effects differ between settlements. Future assessments of CDTI coverage should explicitly consider these interacting factors.

Improving ivermectin coverage among highly mobile populations—particularly small-scale, often informal or illegal miners—requires efforts tailored to their unique challenges. Engaging occupational leaders and aligning MDA schedules with work cycles may further improve access and participation among these underserved groups. We recommend synchronizing MDA delivery with mining activities, potentially through mobile or peer-led distribution at mining sites. Additionally, the use of satellite imagery and GIS, as demonstrated in this study, can enhance the identification of transient settlements and inform targeted planning. We noted unoccupied shelters located within the delineated transmission focal area demonstrating transient settlements. Similar mobile populations, transient settlements, and transmission foci likely exist across the broader Tano–Ankobra and other Operational Transmission Zones (OTZs). Without wider implementation of adapted strategies—such as exhaustive CES and tailored interventions for mobile groups—these areas could pose a risk of reinfection or persistent transmission in regions of Ghana where onchocerciasis elimination is nearly achieved.

The high biting rates recently reported in Kwanware and Ottou—up to 6,000 MBR - reflect a rebound in blackfly populations following the cessation of vector control [15,32]. This, alongside low effective coverage, contributes to an increased force of transmission in affected settlements [15,31]. Similar entomological indices in Togo were historically linked to holoendemicity, with mf prevalence reaching 90% in some sites at baseline. Togo achieved near-elimination through a combination of biannual ivermectin and vector control interventions [55]. Although Ghana implemented vector control during the OCP era, including in rivers near Kwanware and Ottou [31], no recent vector control activities have been reported in this focus. Given the resurgence in blackfly populations and the persistence of transmission despite biannual CDTI, reintroducing focal vector control—such as ground larviciding or the "Slash and Clear" method—may be warranted [15,56]. These strategies, when combined with improved CDTI and monitoring, would accelerate progress toward elimination. Additionally, follow-up entomological and parasitological surveys after several uninterrupted years of biannual CDTI would provide critical insights into transmission trends and the potential need for intensified interventions [57].

Stolk et al.'s modelling study suggests that treatment can be stopped at the 'pixel' level, corresponding to a settlement [20]. However, they acknowledge that such a unit is too small for decision-making. This thinking aligns with findings that the smallest transmission focus spans a 10–12 km radius [15,24], often covering several settlements. Therefore, it appears that the smallest unit for making sustainable decisions is the delineated focus corresponding to the smallest epidemiological unit in onchocerciasis transmission zones. This aligns with the NTD 2030 roadmap and the growing call to localize actions to accelerate onchocerciasis elimination. Investigations, evaluations including CES, and interventions will need to consider this epidemiological reality. Micro-mapping of onchocerciasis transmission residual foci stemming from signal from impact or EoT evaluation is urgently needed to direct effort at this time of funding dilemma [58]. Or else we risk losing the achievement hitherto. The impact of such consideration would even be more assuring and enduring in places where the surrounding areas have received treatment for many years, resulting in a lower/no risk of reinvasion [26], or where there is an adequate buffer zone (>20 km) separating neighbouring foci [42]. Furthermore, model should be developed for elimination of onchocerciasis in delineated transmission focus as suggested by Stolk and colleagues [20]. Effective settlement level geographical coverage, entomological and human dynamics will be important variables to consider in localised elimination modelling.

The main limitations of this study relate to recall bias and social desirability bias. Social desirability may have led to overreporting of ivermectin participation [59], particularly in settlements like Ottou, Kwadom-Joboy and Kwaededen, which reported over 70% effective coverage. This bias is supported by the contradictory community reports regarding the absence of resident CDDs, which often resulted in missed ivermectin doses in these settlements. This suggests that the reported 53.5% effective participation may be an overestimate of the true coverage. On the other hand, recall bias might have led to underreporting [60], as people recall fewer treatments than they received. In one study, recall accuracy was found to be remarkably high (>80%) at 3, 6- and 12-months post-MDA [61], but this is mainly relevant for preceding CDTI (point participation) rather than CDTI rounds further in the past (effective/cumulative participation). To reduce this bias, answer options were categorized. In another study, Breiger et al (2011) reported equal proportions of underestimation and overestimation at 16.2% [22] in their study, meaning the biases cancelled out. This could be the case in this study. However, in the context of over 27 annual rounds of CDTI in our study area [15] and participants being ≥15 years of age, many people would have taken ivermectin well above 10 times and thus are less likely to be incorrectly classified in the "≥10 times" category. All these arguments suggest that the impact of recall bias might be low. Another source of bias is the possible confusion between the number of years of treatment and the number of times of treatment. Biannual treatment started in the study area in 2018, five years earlier, so the number of times might have been different from the number of years. This situation was partially offset by the 2 missed rounds in 2020 due to the COVID-19 pandemic-related suspension of CDTI [62]. Given the mix of annual and biannual rounds in this study area, interpreting participation in terms of the number of rounds is more appropriate. These challenges should be considered in future evaluations, particularly when establishing cut-off points (number times) corresponding to 10 years of treatment.

In all, despite nearly three decades of mass drug administration (MDA) with ivermectin, the Kwanware-Ottou focus in Ghana remains a stubborn hotspot of onchocerciasis transmission, highlighting a critical gap in the country's elimination strategy. This localized persistence—characterized by high microfilariae prevalence and intense blackfly biting rates—underscores the limitations of a uniform approach to achieving elimination. Without intensified interventions and cross-district coordination, such foci risk not only sustaining transmission but also reintroducing infection into areas nearing elimination. Identifying these transmission foci is essential—not only to guide where treatment efforts should be intensified, but also to highlight areas vulnerable to resurgence, which should be prioritized for post-stop MDA and post-elimination surveillance.

We recommend a multi-pronged strategy to address these challenges:

• Developing elimination modelling at the transmission focus level to guide targeted interventions;

• Introducing prospective surveillance systems to monitor individual-level participation in MDA;

- Tailoring MDA delivery to reach mobile and hard-to-reach populations through peer-led distribution and occupational synchronization;

- Reintroducing focal vector control interventions, such as ground larviciding or Slash and Clear.

These measures are essential to safeguard progress toward onchocerciasis elimination in Ghana and elsewhere. As improvements in effective participation may take time to become apparent, we recommend that this exhaustive approach be implemented primarily during the initial investigation of a delineated transmission focus or after at least five rounds/years of CDTI. Including these measures in an exhaustive sampling (census) CES should not cost more than a standard CES conducted at the district level [41], since transmission foci are often much smaller [63,64]. However, formal costing of this approach is needed to better inform national onchocerciasis elimination programmes.

## Conclusion

This study demonstrates inadequate effective participation (i.e., ≥ 10 rounds of ivermectin), often masked by high point coverage, as well as settlement-level suboptimal CDTI coverage within the Kwanware-Ottou onchocerciasis transmission focus. High effective participation was significantly associated with reduced infection. Together with high blackfly biting rates, these low participation levels are sustaining ongoing transmission both within the focus and in surrounding areas. Mobile populations, miners, and residents of remote settlements remain underserved during CDTI. Exhaustive CES sampling in transmission hotspots is essential for identifying coverage gaps and guiding targeted interventions. We recommend a multi-pronged strategy—including focus-level elimination modelling, prospective MDA-participation surveillance, tailored delivery approaches for mobile and hard-to-reach groups, and focal vector control—to safeguard progress toward onchocerciasis elimination in Ghana and beyond.

## Supporting information

**S1 Text. Census Questionnaire.**
(DOCX)

## Acknowledgments

We are grateful to all the participants who willingly took part in this study. Their cooperation and contributions were invaluable. We also extend our sincere appreciation to the programme staff at the National, Regional, District, and Subdistrict levels for their unwavering support and commitment throughout the study. Additionally, we acknowledge the efforts of the community leaders and local stakeholders who facilitated consultations and mobilization, ensuring the smooth execution of the research.

## Author contributions

**Conceptualization:** Rogers Nditanchou, David Agyemang, Mike Yaw Osei-Atweneboana, Richard Selby, Joseph Opare, Louise Hamill, Joseph Nelson Siewe Fodjo, Elena Schmidt, Veronique Verhoeven, Robert Colebunders.

**Data curation:** Rogers Nditanchou, Anita Jeyam, Stephen Pye.

**Formal analysis:** Rogers Nditanchou, Anita Jeyam, Stephen Pye.

**Funding acquisition:** Rogers Nditanchou.

**Investigation:** Rogers Nditanchou, Akinola Stephen Oluwole, Judith Saare, Alexandre Chailloux, Sandra Adelaide King, Mike Yaw Osei-Atweneboana, Joseph Opare, Joseph Nelson Siewe Fodjo.

**Methodology:** Rogers Nditanchou, Judith Saare, David Agyemang, Alexandre Chailloux, Joseph Opare, Anita Jeyam, Louise Hamill, Robert Colebunders.

**Project administration:** Rogers Nditanchou, Judith Saare, David Agyemang, Sandra Adelaide King, Richard Selby, Joseph Opare, Elena Schmidt.

**Resources:** Rogers Nditanchou.

**Supervision:** Rogers Nditanchou, Judith Saare, David Agyemang, Sandra Adelaide King, Richard Selby, Joseph Opare, Joseph Nelson Siewe Fodjo, Veronique Verhoeven, Robert Colebunders.

**Validation:** Rogers Nditanchou, Alexandre Chailloux.

**Visualization:** Rogers Nditanchou, Alexandre Chailloux.

**Writing – original draft:** Rogers Nditanchou.

**Writing – review & editing:** Rogers Nditanchou, Akinola Stephen Oluwole, David Agyemang, Alexandre Chailloux, Sandra Adelaide King, Mike Yaw Osei-Atweneboana, Richard Selby, Joseph Opare, Anita Jeyam, Stephen Pye, Louise Hamill, Joseph Nelson Siewe Fodjo, Elena Schmidt, Veronique Verhoeven, Robert Colebunders.

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
