## [Decision Letter · Decision Letter 0]

15 Jul 2025

Evaluating participation in Ivermectin Mass Drug Administration in the Kwanware-Ottou Persistent Onchocerciasis Transmission Focus in Wenchi, Ghana

Dear Dr. NDITANCHOU,

Thank you for submitting your manuscript to PLOS Neglected Tropical Diseases. After careful consideration, we feel that it has merit but does not fully meet PLOS Neglected Tropical Diseases's publication criteria as it currently stands. Therefore, we invite you to submit a revised version of the manuscript that addresses the points raised during the review process.

Please submit your revised manuscript within 60 days Sep 13 2025 11:59PM. If you will need more time than this to complete your revisions, please reply to this message or contact the journal office at plosntds@plos.org. Please include the following items when submitting your revised manuscript:

We look forward to receiving your revised manuscript.

Kind regards,

Eduardo José Lopes-Torres, Ph.D.

Academic Editor

Jong-Yil Chai

Section Editor

Shaden Kamhawi

co-Editor-in-Chief

Paul Brindley

co-Editor-in-Chief

**Journal Requirements:**

At this stage, the following Authors/Authors require contributions: Rogers NDITANCHOU, Akinola Stephen Oluwole, Judith Sare, David Agyemang, Alexandre Chailloux, Sandra Adelaide King, Mike Yaw Osei-Atweneboana, Richard Selby, Joseph Opare, Anita Jeyam, Stephen Pye, Louise Hamill, Joseph Nelson Siewe Fodjo, Elena Schmidt, Veronique Verhoeven, and Robert Colebunders. Please ensure that the full contributions of each author are acknowledged in the "Add/Edit/Remove Authors" section of our submission form.

4) We do not publish any copyright or trademark symbols that usually accompany proprietary names, eg ©,  ®, or TM  (e.g. next to drug or reagent names). Therefore please remove all instances of trademark/copyright symbols throughout the text, including:

- © on page: 11

- ® on page: 11.

5) Please upload all main figures as separate Figure files in .tif or .eps format. For more information about how to convert and format your figure files please see our guidelines:

6) We have noticed that you referred to (Appendix 1) in your manuscript on pages 8 and 9. However, there is no corresponding file uploaded to the submission. Please upload it as a separate file with the item type 'Supporting Information'. Please add a full list of legends for your Supporting Information files after the references list.

7)  Some material included in your submission may be copyrighted. According to PLOSu2019s copyright policy, authors who use figures or other material (e.g., graphics, clipart, maps) from another author or copyright holder must demonstrate or obtain permission to publish this material under the Creative Commons Attribution 4.0 International (CC BY 4.0) License used by PLOS journals. Please closely review the details of PLOSu2019s copyright requirements here: PLOS Licenses and Copyright. If you need to request permissions from a copyright holder, you may use PLOS's Copyright Content Permission form.

Potential Copyright Issues:

i) Figures 1. Please (a) provide a direct link to the base layer of the map (i.e., the country or region border shape) and ensure this is also included in the figure legend; and (b) provide a link to the terms of use / license information for the base layer image or shapefile. We cannot publish proprietary or copyrighted maps (e.g. Google Maps, Mapquest) and the terms of use for your map base layer must be compatible with our CC BY 4.0 license.

ii) Figure 2: Thank you for stating "river network from HydroSheds; https://www.hydrosheds.org/products/hydrosheds." We noted that the license is not compatible with our CC BY 4.0 License. Please amend the relevant section of the figure to be from a source that is compatible with our CC BY 4.0 License.

8) We note that your Data Availability Statement is currently as follows: "All relevant data are within the manuscript and its Supporting Information files;" however, there are not any supporting information files uploaded to the submission.

9) Please amend your detailed Financial Disclosure statement. This is published with the article. It must therefore be completed in full sentences and contain the exact wording you wish to be published.

3) If any authors received a salary from any of your funders, please state which authors and which funders.

Please ensure that the funders and grant numbers match between the Financial Disclosure field and the Funding Information tab in your submission form. Note that the funders must be provided in the same order in both places as well.

**Reviewers' Comments:**

Reviewer's Responses to Questions

**Key Review Criteria Required for Acceptance?**

**Methods**

-Are the objectives of the study clearly articulated with a clear testable hypothesis stated?

-Is the study design appropriate to address the stated objectives?

-Is the population clearly described and appropriate for the hypothesis being tested?

-Is the sample size sufficient to ensure adequate power to address the hypothesis being tested?

-Were correct statistical analysis used to support conclusions?

-Are there concerns about ethical or regulatory requirements being met?

Reviewer #1: The objectives and hypothesis are clear for this study. The data used is partially historical: microfilarial prevalence levels from 1989 and mf and OV16 levels from 2021, and partially self reported by participants surveyed for effective participation. The authors acknowledge that this could potentially result in over reporting in some instances (social desireability), and underreporting in others due to recall bias, which means they could cancel each other out. The statistical methods are appropriate for the data, thogh some sites are poorly sampled, so it might be beneficial to redo the regression model with only well sampled sites to compare odds ratios. I have no ethical or regulatory concerns

Reviewer #2: (No Response)

Reviewer #3: Lines 115-124: Is there baseline information on onchocerciasis endemicity or the pre-control prevalence (%) in Wenchi HD? While I noticed that some data are later presented in Table 2, it may be helpful to briefly mention the baseline status here for context. Additionally, could the authors clarify how the COVID-19 pandemic affected programme implementation? Was uninterrupted biannual CDTI maintained since 2018, or were there any disruptions? If there was a temporary halt, I recommend discussing its potential implications for onchocerciasis control and elimination progress in both this focus and in Ghana overall in the Discussion section (e.g., see paper "What does the COVID-19 pandemic mean for the next decade of onchocerciasis control and elimination?").

Line 123: I appreciate that confidence intervals (CIs) are provided for microfilarial prevalence. I encourage the authors to also report CIs for other key measures in this paragraph, including blackfly infectivity rates if possible. If CIs are not feasible, please consider indicating the sample sizes (e.g., how many individuals or flies were surveyed), as this would help readers assess representativeness.

Line 124: Please replace “5.9‰” with “5.9%” and remove the unnecessary period at the end of the sentence. I also recommend defining abbreviations such as “km” and “CI” upon their first use.

Lines 125-127: Given that the transmission focus spans three HDs, were there any periods when different control interventions were implemented in these HDs, or did all three consistently receive annual/biannual CDTI simultaneously? (and vector control?)

Figure 1: I suggest including, perhaps in the figure subtitle, the years during which serological and parasitological surveys were conducted, as well as the years of fly collection and larval inspection. In the Methods section, it may be informative to cite the range of monthly biting rates or MTP from reference 20, as these data help contextualise the transmission potential. The reported MBRs from reference 20 (5,000–6,000 in Ottu and Kwanware) seem high relative to the moderate baseline endemicity in Table 2. Were the MBRs always this high, or have they fluctuated? Some clarification on the relationship between these metrics would be helpful.

Indeed, given the persistent transmission and high MBRs, in contrast to the moderate baseline microfilarial prevalence of this focus (e.g., hypo- to mesoendemic), would it be more relevant to explore which community groups may be exposed to the high MBRs (subgroup of the population may be considered hyperendemic at baseline).

Lines 140-142: How was the term “transmission focus” defined? Did the authors follow a specific criterion (e.g., WHO’s 10 km guideline) for this designation?

Lines 145-147: The use of GPS satellite imagery to identify settlements and its integration with field and government data is somewhat innovative, commendable methodology.

Lines 177-178: While I understand the rationale for capping reported MDA participation at 10 rounds, especially since biannual treatment began only in 2018, I suggest clarifying that this may not perfectly reflect years of treatment. The current discussion appropriately notes that 10 rounds may not always correspond to 10 years.

Lines 179-182: The classification of “non-effective” participation as fewer than 10 rounds may be affected by recall and desirability biases. I recommend considering exploratory analyses using shorter thresholds, ideally based on prior literature and your own data collection to further strengthen this aspect.

Line 206: Should “Ov16” read “anti-Ov16”? Were participants tested for antibodies to the Ov16 antigen (anti-Ov16), or for the antigen itself (Ov16)? Please clarify and use consistent terminology throughout the manuscript.

Line 207: When referring to “communities,” is this synonymous with the 19 “settlements” described in the Results? Please ensure consistent terminology to aid reader understanding.

Length of stay in the community: For participants with up to 10 years of residence, were responses capped or excluded if they had not lived in the area for the full duration of MDA implementation? Alternatively, was an exploratory analysis conducted to assess whether these individuals influenced the results? Clarification here would improve the robustness of the findings.

**Results**

-Does the analysis presented match the analysis plan?

-Are the results clearly and completely presented?

-Are the figures (Tables, Images) of sufficient quality for clarity?

Reviewer #1: Yes.

The figures are sufficient for the analysis, and clear.

Reviewer #2: (No Response)

Reviewer #3: Line 226: Was age assessed for normality before reporting the mean and standard deviation (SD)? Given the inclusion criteria starts at age 15, the age distribution may be skewed, so median and interquartile range (IQR) might be more appropriate. Please define “SD” as “standard deviation” upon first use.

Line 228: Table 1 does not require brackets. Additionally, please check for typographical errors such as “participati on” and “co verage,” which appear in my version of the manuscript.

Table 1: Table 1 addresses two important concepts: effective participation (at least 10 rounds of ivermectin) and point participation (participation in the last round). Many programmes emphasise point participation, but effective participation is often more critical for infection clearance. While recall bias limits this measure’s precision, the distinction is valuable; point participation is relatively high (80.3%) among the eligible population, but effective participation is notably lower (53.4%).

Table 2: It is notable that this hypo- to mesoendemic focus at baseline continues to exhibit substantial onchocerciasis transmission (as shown by microfilariae and anti-Ov16 prevalence, 0–30% and 1–19%, respectively); but again, perhaps less suprising given the very high MBRs. Are there particular communities within the area, or subgroups (e.g., those closer to blackfly breeding sites with >400 larvae found, per Figure 1), that have disproportionally higher transmission or disease burden? If so, could this inform a more targeted approach to MDA or surveillance? If data in Table 2 were sourced from Reference 20, please include a citation in the table.

Additionally, what does the hyphen “–” signify in Table 2? Does it indicate missing data? For the Branam community at baseline, the cell is blank, could this be clarified? Further, the methodology for assessing microfilariae and anti-Ov16 prevalence is not described in detail. Was microfilarial prevalence assessed in adults? How were skin snips performed (e.g., two samples per person, inverted or classic microscopy)? Which assay was used for anti-Ov16? If these results are to remain in the Results, I suggest adding this methodological detail in the Methods section.

Lastly, why is the Ottu community not mentioned in Table 2, given the focus on the Kwanware-Ottu persistent transmission focus? Reference 20 appears to have data for both microfilarial prevalence and anti-Ov16 seroprevalence for Ottu.

Line 241: Is there a specific reason for enumerating this paragraph as “1.” without subsequent numbered items? If not, the enumeration can be removed for clarity.

Table 3: Were variables assessed for multicollinearity, particularly between ethnic groups and sub-districts, since certain sub-districts may predominantly comprise particular ethnic groups or occupations (e.g., miners in Boase)? The Discussion (lines 323–325) notes that sub-districts may overlap substantially with occupational groups.

Lines 247–256: Please provide complete comparisons in the text. For instance, “miners and students or other occupations have significantly lower odds of being in the higher categories” should specify “compared to farmers.” Please clarify the reference categories in sentences about ethnic groups and sub-districts as well.

Line 262: Instead of a standalone sentence, please include “(see Table 4)” in brackets.

Lines 260–264: The phrase “associations in the same direction” would benefit from clarification. This likely means that the direction of associations for effective and point participation is consistent. Additionally, how does this compare to the previous model, given that the explanatory variables are the same?

Lines 261–262: Is there a reason for the observed differences between effective and point participation models, particularly regarding the “travel category”? Additional explanation would be valuable.

**Conclusions**

-Are the conclusions supported by the data presented?

-Are the limitations of analysis clearly described?

-Do the authors discuss how these data can be helpful to advance our understanding of the topic under study?

-Is public health relevance addressed?

Reviewer #1: This manuscript is of public health interest, though additional details on how to further understand this data could be presented. Descriptive methods could be used to explain how the anomalous Ottou site, with very high rates of effective and point participation differs from the other sites.

Reviewer #2: (No Response)

Reviewer #3: Lines 273–274: The statement, “The exhaustive sampling provided accurate and precise estimates, with negligible sampling error (24),” implies formal testing for accuracy and precision. If such testing was performed, please add the relevant details to the Methods and Results. If not, I suggest revising to a more tentative statement, such as, “The exhaustive sampling aimed to provide accurate and precise estimates…”

Line 279: Rather than “some,” please specify the exact number of settlements (e.g., “ten”).

Lines 283–285: The phrase “would have been missed during usual CES” suggests certainty. If this is hypothetical, please rephrase to “could have been missed…” or provide supporting evidence.

Lines 303–304: Clarify how “indicates a more stable model when changing from the treatment category 5–9 times to ≥10 times” relates to the earlier statement in lines 255–256 (“The model is most stable for change from 2 (5–10 times) to 3 (>10 times) categories”). The intervals are defined differently; please ensure consistency or explain the distinction.

Lines 307–308: Could you clarify why “<10 times” and “≥10 times” is a viable cut-off for effective participation? While this threshold aligns with programmatic guidance, the model explains only 14.5% of the variability, which may represent a normal range for community settings but still relatively low, especially when compared to the point model’s 34.2%.

Lines 328–329:

Do you have specific recommendations for improving coverage among highly mobile communities (e.g., miners, farmers)? Would strategies such as delivering ivermectin during mining activities, or incorporating satellite data for planning, help improve treatment coverage?

Lines 332–342: I commend the thorough limitations section, particularly the discussion of recall and social desirability biases. However, in lines 340–342, caution is needed: while some individuals may have received more than 10 treatments over three decades, interruptions could allow adult worms to resume microfilariae production and sustain transmission. The impact of missing one or two rounds, whether consecutively or intermittently over 10 years, may differ and could be explored further.

Vector Dynamics and Control Strategies: Given the high biting rates observed (up to 6,000 MBR, per Reference 20), I encourage the authors to discuss their implications. In Togo, such rates were associated with holoendemicity (up to 90% microfilariae prevalence at baseline), yet elimination was nearly achieved through a combination of biannual ivermectin and vector control (see “Reaching Elimination of Onchocerciasis Transmission with Long-term Vector Control…” and “Impact of ivermectin and vector control on onchocerciasis transmission in Togo…”). Reference 20 notes that Ghana also implemented vector control during OCP, including in rivers near Kwanware and Ottou, but vector control is not mentioned in the manuscript. Would the authors recommend re-implementing focal vector control (e.g., ground larviciding or Slash and Clear), alongside improved CDTI, for persistent foci? Might further surveys soon after multiple years of biannual CDTI provide greater insight into trends?

Policy Recommendations: A brief policy recommendation paragraph before the conclusion would be helpful. Besides using elimination modelling, do the authors suggest introducing prospective surveillance for effective participation to reduce recall bias, or other surveillance improvements? As well as some of the recommendations I suggsted earlier.

Mobility and Reinfection Risk: In lines 215–216, it is noted that “many unoccupied temporal shelters” are used by farmers, miners and cattle rearers. Do these groups reside within or outside the transmission focus? If outside, could they act as a source of reinfection to areas in Ghana where elimination is nearly achieved?

Contextualisation: How does the Kwanware-Ottou focus fit within the broader context of onchocerciasis control and elimination efforts in Ghana?

**Editorial and Data Presentation Modifications?**

Reviewer #1: It appears that Dagaaba is misspelled as Dagaarba throughout the manuscript. Please ensure that this spelling is appropriate and not a lesser used or historical name for this group.

Reviewer #2: (No Response)

Reviewer #3: Background Section

Line 68: Consider whether the abbreviation “Ov” is necessary. In lines 76–77, you use the standard abbreviation “O. volvulus,” which is more common in the literature.

Lines 69–70: It would strengthen your argument to add a reference for the onchocerciasis burden. Additionally, consider mentioning that the burden of onchocerciasis has recently been associated with neurological complications, such as onchocerciasis-associated epilepsy.

Lines 75–76: Please clarify whether “therapeutic coverage” refers to the overall population or to the eligible population (likely the latter, given the scope of the manuscript).

Lines 74–77: The explanation that ivermectin must be administered for several years to match the reproductive lifespan of the adult worm is helpful. For clarity, please add that ivermectin is microfilaricidal, it kills the larvae (microfilariae) of O. volvulus, not the adult worms, therefore the long treatment. Also, the 85% coverage threshold for eligible population is typically cited for the onchocerciasis elimination programme in the Americas, while 80% is the standard for Africa (see Guidelines for stopping mass drug administration and verifying elimination of human onchocerciasis: criteria and procedures). Is there a reason for using 85% as the benchmark for Ghana in this article? If so, please clarify. That said; I agree that higher coverage is always preferable, especially in such a high-transmission focus.

Lines 81–82: When stating, “To assess this, it is recommended that coverage evaluation surveys (CES) be conducted every 1–3 years at the evaluation unit level,” please specify who recommends this and include a reference.

Lines 84–85: These lines, while clearer, appear to partially repeat information from lines 80–81. Consider consolidating for clarity and conciseness.

Lines 87–88: Although APOC will be familiar to expert readers, it would be helpful in earlier paragraphs, when introducing onchocerciasis control, EOT, Ghana, etc., to mention that onchocerciasis control in Africa began with OCP and later APOC.

Line 89: Regarding the statement, “little consideration has been given to this kind of evaluation thereafter (13),” please clarify what “thereafter” refers to. Does it mean after a programme in the APOC area reaches 8 years of MDA, or after a specific year when such surveys were mainly conducted?

Lines 91–93: Since geographic and therapeutic coverage are key components of the manuscript, I recommend briefly defining how each is determined in the introduction.

**Summary and General Comments**

Reviewer #1: The manuscript “Evaluating participation in Ivermectin Mass Drug Administration in the Kwanware-Ottou Persistent Onchocerciasis Transmission Focus in Wenchi, Ghana” discusses current shortfalls in ivermectin MDA efforts in treating Onchocerciasis, a neglected but very treatable filarial parasite in the Wenchi health district. Both historical data (microfilarial prevalence levels from 1989 and mf and OV16 levels from 2021) are compared against survey data on usage of more than 10 administrations of ivermectin (effective), compared against ivermectin administration in the most recent round of CDTI.

Overall the manuscript is well written and the statistical methods are appropriate for the types of data used in the study. The results should be of interest to those in public health and those in the filarial community

I have only minor comments:

- Curiously, some regions exceed the recommended levels of effective participation in MDA (Ottou, Kwadom Joboy). In the case of Ottou it is excluded from the analysis as an outlier. It nevertheless stands as an anomaly. Based on the regression model it would still be interesting to see how Ottou fits the model.

- It appears that Dagaaba is misspelled as Dagaarba throughout the manuscript. Please ensure that this spelling is appropriate and not a lesser used or historical name for this group.

Reviewer #2: The authors evaluated participation in ivermectin mass drug administration (MDA) for onchocerciasis in the Wenchi district of Ghana. The study highlights that low participation rates and limited geographical (settlement-level) coverage contribute to persistent transmission of the disease.

1. The manuscript focuses solely on participation, but does not provide any information on compliance, which is essential to fully assess the effectiveness of CDTI efforts.

2. Annex 1 is referenced but not included in the submission.

3. The manuscript lacks a broader discussion comparing findings with those from other countries, and especially from other districts or regions within Ghana. Several studies have been published on participation and compliance with CDTI, and integrating these comparisons would help identify key lessons to improve program success and support onchocerciasis elimination.

4. Figure 1: An overview map of Ghana should be included before the study area map to help readers understand the location of the Wenchi district within the national context.

5. Table 1: The asterisk (*) next to "Ottou" needs to be explained in a footnote or legend for clarity.

6. In the Results section, it would be helpful to briefly summarize the key findings from the tables in one to two sentences. This would allow readers to quickly grasp the main take-home messages.

Reviewer #3: This is an interesting and timely paper, particularly relevant as sub-Saharan Africa advances toward the ambitious goal of onchocerciasis transmission elimination and eventual eradication. I appreciate that the authors provided a version of the manuscript with numbered lines, which facilitated referencing specific points throughout this review.

One of the main strengths of the paper lies in its thoughtful discussion of the limitations inherent in reporting geographic and therapeutic coverage at the district or larger administrative levels, an issue that often results in missing small settlements that may act as persistent pockets of transmission. The authors also draw attention to the significant impact of intermittent ivermectin intake: missing several years of treatment may prevent infection clearance, a crucial and sometimes underappreciated challenge in longstanding onchocerciasis control efforts across sub-Saharan Africa. These points are known but relevant, especially in the context of a high-endemicity area in Ghana where nearly three decades of ivermectin distribution have not interrupted transmission.

Overall, the manuscript is comprehensive and demonstrates critical thinking, offering a wealth of information on the transmission focus under study. The inclusion of detailed epidemiological and programmatic data, along with reflections on the challenges and implications for control, is commendable and adds value to the field. Such thoroughness is not always present in programmatic literature and will be appreciated by both researchers and policy-makers, especially after answering some of review queries for complementary information.

The study’s key strengths include its exhaustive approach to data collection, the integration of both epidemiological and entomological findings, and the critical reflection on programmatic and methodological limitations. The discussion is balanced and the policy recommendations are well founded, although they could be further expanded, especially regarding strategies to address persistent transmission in hard-to-reach communities and within Ghana.

Some minor points, such as the need for more consistent terminology, clarification of methods, and a more cautious interpretation of some outcomes, are addressed in detail in the specific comments. While part of the data has been publsihed before, there are no concerns about research ethics or dual publication. Overall, I believe the manuscript makes a significant contribution to the understanding of onchocerciasis control and surveillance in highly endemic, resource-constrained settings of West Africa.

I recommend publication after revision, with attention to the specific methodological clarifications and policy recommendations outlined in my comments.

PLOS authors have the option to publish the peer review history of their article (what does this mean? ). If published, this will include your full peer review and any attached files.

**Do you want your identity to be public for this peer review?** For information about this choice, including consent withdrawal, please see our Privacy Policy .

Reviewer #1: No

Reviewer #2: No

Reviewer #3: **Yes:** Luís-Jorge Amaral

**Figure resubmission:**

**Reproducibility:**



---

## [Decision Letter · Decision Letter 1]

2 Dec 2025

Evaluating participation in Ivermectin Mass Drug Administration in the Kwanware-Ottou Persistent Onchocerciasis Transmission Focus, Wenchi, Ghana

Dear Dr. NDITANCHOU,

Thank you for submitting your manuscript to PLOS Neglected Tropical Diseases. After careful consideration, we feel that it has merit but does not fully meet PLOS Neglected Tropical Diseases's publication criteria as it currently stands. Therefore, we invite you to submit a revised version of the manuscript that addresses the points raised during the review process.

* A rebuttal letter that responds to each point raised by the editor and reviewer(s). You should upload this letter as a separate file labeled 'Response to Reviewers '. This file does not need to include responses to any formatting updates and technical items listed in the 'Journal Requirements' section below.

* A marked-up copy of your manuscript that highlights changes made to the original version. You should upload this as a separate file labeled 'Revised Manuscript with Track Changes '.

* An unmarked version of your revised paper without tracked changes. You should upload this as a separate file labeled 'Manuscript '.

We look forward to receiving your revised manuscript.

Kind regards,

Eduardo José Lopes-Torres, Ph.D.

Academic Editor

Jong-Yil Chai

Section Editor

Shaden Kamhawi

co-Editor-in-Chief

Paul Brindley

co-Editor-in-Chief

**Journal Requirements:**

1) Please amend your detailed Financial Disclosure statement. This is published with the article. It must therefore be completed in full sentences and contain the exact wording you wish to be published.

2)  Please ensure that the funders and grant numbers match between the Financial Disclosure field and the Funding Information tab in your submission form. Note that the funders must be provided in the same order in both places as well.

**Reviewers' comments:**

Reviewer's Responses to Questions

**Key Review Criteria Required for Acceptance?**

**Methods**

-Are the objectives of the study clearly articulated with a clear testable hypothesis stated?

-Is the study design appropriate to address the stated objectives?

-Is the population clearly described and appropriate for the hypothesis being tested?

-Is the sample size sufficient to ensure adequate power to address the hypothesis being tested?

-Were correct statistical analysis used to support conclusions?

-Are there concerns about ethical or regulatory requirements being met?

Reviewer #2: (No Response)

Reviewer #3: Methods:

1) Line 154: “Microfilariae” does not need a hyphen in the manuscript text.

2) Table 1 and related text: It would be helpful to clarify when vector control started in this health district. Was it in nineteen seventy four, as in Ghana overall If so, the prevalence reported for nineteen eighty would not correspond to a true baseline, which suggests hyperendemicity and high transmission potential before control. This clarification is mentioned later around paragraph 170, but it would be very useful to state it earlier in the table legend or in the first methods paragraph with the epidemiological indicators.

3) First methods paragraph and Table 1: Just an observation, the increasing trend in prevalence in Kwanware and Ottou, although based on recent small sample sizes, suggests high transmission potential, compounded by sub-optimal effective CDTI as you discuss in the manuscript. It could be interesting to conduct a brief human landing catch survey in this location.

Figure 1 is very clear and visually informative.

4) Line 184: It is not certain that all readers will immediately recognise the meaning of “%0” notation. I would consider briefly defining it the first time it appears.

**Results**

-Does the analysis presented match the analysis plan?

-Are the results clearly and completely presented?

-Are the figures (Tables, Images) of sufficient quality for clarity?

Reviewer #2: (No Response)

Reviewer #3: Results:

Figure 2 is also clearly presented and easy to interpret.

1) Table 2: For Ottou, please consider reporting the exact p-value if possible. At present it is written as “less than zero point two four three” or similar. The "<" may just be a typo.

2) Lines 305 and 306: You state that the young population had a coverage of less than fifty five percent. If available, please provide the precise percentage.

3) Line 327: Consider whether “microfilaria prevalence” should be “microfilarial prevalence” for consistency of terminology.

4) Lines 307 and 322 Ottou as an outlier: I am not entirely convinced that Ottou should be excluded from the analyses as an outlier. While the sample size is indeed small and neighbouring settlements have lower prevalence, this needs careful consideration. Either

a) the population in Ottou truly has high point and longitudinal participation in ivermectin due to their location or knowledge of its importance, and prevalence is driven by high biting rates combined with lower coverage in nearby settlements, or

b) the method used may be less reliable when sample sizes are very small for example with only nineteen individuals, due to biases such as recal bias.

It is likely that both mechanisms contribute, although the first may be particularly important. There may also be additional, less obvious reasons. Importantly, onchocerciasis transmission is highly heterogeneous, in contrast to infections such as influenza where exposure is more homogeneous. Some settlements often experience much higher blackfly biting rates. For this reason I would still consider including Ottou in the analyses, perhaps with an appropriate note of caution.

5) Table 3 participation: Please check the proportions for cumulative participation. Several entries appear to exceed one hundred percent when summed by row. For instance 21 participants out of 52 is not 65.3%, and 121 out of 353 is not 27.1%.

6) Table 3 microfilarial prevalence and anti-Ov16: For the values given in brackets for microfilarial prevalence and anti-Ov16, please specify explicitly in the Table what these represent for example 95% confidence intervals.

7) Table 4: “Cattle rearing/” can be simplified by removing the slash?

8) Given the abundance of rivers in the study area, fishing is mentioned briefly around line 209. If fishing is a relevant occupation in this population, please clarify whether it was considered as a variable and if not (or no one reported the profession?), whether this potential heterogeneity in exposure is adequately captured under related high-risk occupations such as farming.

**Conclusions**

-Are the conclusions supported by the data presented?

-Are the limitations of analysis clearly described?

-Do the authors discuss how these data can be helpful to advance our understanding of the topic under study?

-Is public health relevance addressed?

Reviewer #2: (No Response)

Reviewer #3: Discussion:

1) Lines 375 to 377: I would be cautious about emphasising the coverage estimates from the 19 settlements too strongly, since some settlements are very small and the estimates may be sensitive to a few households being missed or not targeted. This is more of a nuance than a requested change, as the current text is very acceptable, and indeed the authors express they view well of going into detailed data of onchocerciasis exposure and control.

2) Please define the acronym CDD the first time it appears on the main text.

3) Suboptimal coverage: One question that remains is whether suboptimal coverage in some settlements is mainly due to a lack of willingness to take ivermectin at that round, to people being absent or working when distributors visited, or to CDDs moving through too quickly and missing households. A short reflection on these potential mechanisms would add useful context. I might have missed it in the discussion?

4) Lines 412 and 413: A reference could be needed to support the statement in these lines.

5) Closing remark: Overall, the manuscript is a clear and valuable account of Ghana and West Africa efforts to eliminate onchocerciasis while addressing persistent transmission. The analysis of effective participation is particularly informative and underlines the importance of heterogeneity in ivermectin uptake, in parallel with heterogeneous exposure to blackfly bites. The attention to mobile populations is also timely and relevant, since these groups may increasingly act as reservoirs and threaten elimination goals.

**Editorial and Data Presentation Modifications?**

Reviewer #2: (No Response)

Reviewer #3: Abstract:

1) Line 35: Please consider also reporting confidence intervals for the main quantitative results, especially for effective participation coverage.

Author summary

2) Line 47: The expression “onchocerciasis public health challenge” resembles “public health problem”, which accurately describes the situation in West Africa before OCP. Since onchocerciasis-related morbidity, such as blindness, is now mostly suppressed in Ghana at the moment, it might be more precise to say that “transmission persists in parts of Ghana” or similar wording.

3) Line 48: There is a missing full stop at the end of the sentence. Otherwise, the author summary reads clearly and is easy to follow.

Background:

4) Line 83: A reference appears to be missing for the statement in this line.

5) Lines 85 and 86: While it is correct that OCP initially relied on vector control, ivermectin delivery became very important from the early nineteen nineties, in some areas even replacing vector control, still during OCP. Therefore, as you explain later in the manuscript, OCP also relied on ivermectin in its later phases. The sentence would benefit from reflecting this evolution more explicitly.

6) Lines 91 and 92: This sentence could be strengthened by citing the relevant APOC report that documents elimination as a public health problem being achieved.

7) Lines 106 and 107: A reference is needed for the reproductive lifespan of the parasite. Otherwise, this paragraph now reads well and is informative.

**Summary and General Comments**

Reviewer #2: The authors have addressed all raised questions and revised the manuscript accordingly. I therefore support publication.

Reviewer #3: The revised manuscript reads as a very informative contribution and a strong testament to the long term efforts in Ghana and West Africa to eliminate onchocerciasis despite persistent transmission in some areas. The methodology used to quantify effective participation is particularly insightful and highlights important heterogeneity in ivermectin uptake, which is meaningful given that exposure to onchocerciasis is also heterogeneous. The discussion of mobile populations is especially relevant, as these groups may increasingly act as reservoirs and jeopardise elimination efforts as transmission becomes more focal.

PLOS authors have the option to publish the peer review history of their article (what does this mean? ). If published, this will include your full peer review and any attached files.

**Do you want your identity to be public for this peer review?** For information about this choice, including consent withdrawal, please see our Privacy Policy .

Reviewer #2: No

Reviewer #3: **Yes:** Luís-Jorge Amaral

**Figure resubmission:**

**Reproducibility:** To enhance the reproducibility of your results, we recommend that authors of applicable studies deposit laboratory protocols in protocols.io, where a protocol can be assigned its own identifier (DOI) such that it can be cited independently in the future. Additionally, PLOS ONE offers an option to publish peer-reviewed clinical study protocols. Read more information on sharing protocols at https://plos.org/protocols?utm_medium=editorial-email&utm_source=authorletters&utm_campaign=protocols

---

## [Editor Report · Decision Letter 2]

18 Jan 2026

Dear Dr. NDITANCHOU,

Thank you for submitting your manuscript to PLOS Neglected Tropical Diseases. After careful consideration, we feel that it has merit but does not fully meet PLOS Neglected Tropical Diseases's publication criteria as it currently stands. Therefore, we invite you to submit a revised version of the manuscript that addresses the points raised during the review process.

* A letter that responds to each point raised by the editor and reviewer(s). You should upload this letter as a separate file labeled 'Response to Reviewers '. This file does not need to include responses to any formatting updates and technical items listed in the 'Journal Requirements' section below.

* A marked-up copy of your manuscript that highlights changes made to the original version. You should upload this as a separate file labeled 'Revised Manuscript with Track Changes '.

* An unmarked version of your revised paper without tracked changes. You should upload this as a separate file labeled 'Manuscript '.

We look forward to receiving your revised manuscript.

Kind regards,

Eduardo José Lopes-Torres, Ph.D.

Academic Editor

Jong-Yil Chai

Section Editor

Shaden Kamhawi

co-Editor-in-Chief

Paul Brindley

co-Editor-in-Chief

**Journal Requirements:**

1)  Please ensure that the funders and grant numbers match between the Financial Disclosure field and the Funding Information tab in your submission form. Note that the funders must be provided in the same order in both places as well.

**Reviewers' comments:**

**Figure resubmission:**
---

## [Editor Report · Decision Letter 3]

28 Jan 2026

Dear NDITANCHOU,

We are pleased to inform you that your manuscript 'Evaluating participation in Ivermectin Mass Drug Administration in the Kwanware-Ottou Persistent Onchocerciasis Transmission Focus, Wenchi, Ghana' has been provisionally accepted for publication in PLOS Neglected Tropical Diseases.

Best regards,

Eduardo José Lopes-Torres, Ph.D.

Academic Editor

Jong-Yil Chai

Section Editor

Shaden Kamhawi

co-Editor-in-Chief

Paul Brindley

co-Editor-in-Chief

---

## [Editor Report · Acceptance letter]

Dear NDITANCHOU,

We are delighted to inform you that your manuscript, "Point Participation Does Not Reflect Long‑Term Treatment Adherence: An Evaluation of Ivermectin MDA in the Kwanware‑Ottou Persistent Onchocerciasis Transmission Focus, Wen-chi, Ghana," has been formally accepted for publication in PLOS Neglected Tropical Diseases.

Best regards,

Shaden Kamhawi

co-Editor-in-Chief

Paul Brindley

co-Editor-in-Chief
